# From black hole interior to quantum complexity through operator rank

Alexey Milekhin

Institute for Quantum Information and Matter, California Institute of Technology, Pasadena,
CA 91125, USA

## Abstract

It has been conjectured that the size of the black hole interior captures the quantum gate complexity of the underlying boundary evolution. In this short note we aim to provide a further microscopic evidence for this by directly relating the area of a certain codimension-two surface traversing the interior to the depth of the quantum circuit. Our arguments are based on establishing such relation rigorously at early times using the notion of operator Schmidt rank and then extrapolating it to later times by mapping bulk surfaces to cuts in the circuit representation.

## 1 Motivation

In the realm of quantum dynamics, one very interesting quantity is the circuit complexity: the minimal number of elementary gates required to build a given unitary. Sometime ago it was suggested [1–4] that the "size" of black hole interior is equal to the complexity of the circuit preparing that black hole state. Here we used "size" in the quotation marks because there are many different proposals of what this geometric quantity should be: volume [1, 2, 5, 6], action inside the Wheeler–DeWitt patch [3, 4, 7], more generic functionals [8, 9]. These proposals pass many interesting *indirect* checks: initial linear growth, switchback effect [10] and even saturation at exponentially late times [11]. However, circuit complexity is exceptionally hard to compute [a]. Moreover, it depends on the set of basis gates and there might be several circuits which are locally minimal. More progress can be obtained in $1 + 1$ bulk dimensions and the Sachdev–Ye–Kitaev (SYK) model [13–16] by directly mapping [17–19] the chord representation [20, 21] to Krylov complexity [22–29]. Thus, unfortunately, most known checks of this proposal are indirect. It is tantalizing to find a *proxy* for complexity which is computable and for which it is possible to establish a general *direct* link with some *concrete* geometric quantity in the interior:

$$\text{A circuit complexity proxy} \xleftrightarrow{\text{direct}} \text{A concrete geometric quantity.}$$

---

[a]We refer to [12] for a recent progress in this direction.

*The purpose of this short note is to present such a connection. Namely, we will argue from the first principles that the area of a certain extremal codimension-two surface passing through the black hole interior, namely Hartman–Maldacena (HM) surface [30], bounds from below the circuit depth of the boundary evolution operator:*

$$\boxed{\frac{\mathscr{A}_{HM}}{4G_N} \leq C \times (\text{circuit depth})}, \tag{1}$$

where $C = \mathscr{A}_\partial (\log D) r^2 / 4$ is a time-independent constant: $D$ is the local Hilbert space dimension, $r$ is related to the connectivity of the underlying quantum circuit and $\mathscr{A}_\partial$ is the boundary area of the region the HM surface is anchored to (in fact, there is the same factor in $\mathscr{A}_{HM}$). For early times we will be able to establish the inequality (1) rigorously. For late times it is a conjecture which we will back up by explicitly mapping a gravity computation to a quantum circuit picture. Circuit depth is a measure of how complicated the underlying circuit is: how many layers of gates are needed to prepare it. So it is similar to circuit complexity. In the translationally-invariant case they are proportional to each other: total number of gates equals depth times the system's size.

*It is important that in eq. (1), the circuit depth is any circuit depth which prepares a given unitary. This relation will allow us to clarify the result of gravitational computation. We will argue that it indeed probes a global minimum and that, surprisingly, the set of basis gates can be chosen arbitrary.*

The main idea of this paper is to interpret the two sides of the thermofield-double (TFD) state not as two independent systems, but the past and the future of the same system. Then entanglement entropy in this state can be viewed as "entanglement in time". We will obtain the inequality (1) by invoking an auxiliary quantity – the operator Schmidt rank $\chi$:

$$\text{Early times}: \frac{\mathscr{A}_{HM}}{4G_N} \leq \log(\chi) \text{ and } \log(\chi) \leq C \times (\text{circuit depth}). \tag{2}$$
$$\text{\small(rigorous)}$$

The operator rank $\chi$ is a feature of given unitary only, it does not depend on the choice of gate set. At late times (beyond the total system's size) $\log(\chi)$ saturates to a constant value. However, the HM surface continues to exist and to grow at the same rate. The same is expected for the circuit depth. Moreover, we will be able to directly translate the bulk Ryu–Takayanagi (RT) [31, 32] surfaces to minimal cuts in the circuit representation, where the HM surface maps to the temporal cut probing the depth. This is why we conjecture inequality (1).

The two regimes and the behavior of all three quantities is illustrated by Figure 1. In case of an infinite system, the late time regime is never realized. To make a stronger connection between operator rank and complexity, in Appendix A we will argue that $\log \chi$ satisfies the properties of complexity, such as subadditivity and switch-back effect.

Usually obtaining lower bounds on complexity (in the broad sense of this words) is hard, this is why the inequality (1) is interesting by itself.

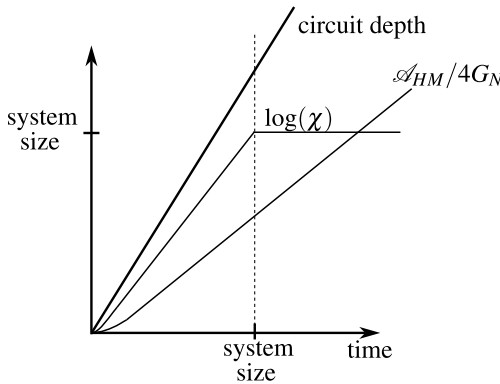

Figure 1: The sketch of the behavior of various quantities as a function of time. We expect that all three have the same linear slope at early times, but we cannot show this rigorously.

## 2 Derivation

Given two copies (left $L$ and right $R$) of a system, we can defined time-evolved TFD state $|\text{TFD(t)}\rangle$ as:

$$|\text{TFD(t)}\rangle = \frac{1}{\sqrt{Z}}\sum_n e^{(-\beta/2-it)E_n}|n\rangle_L|n\rangle_R, \tag{3}$$

where $|n\rangle$ are energy eigenstates and $Z = \sum_n e^{-\beta E_n}$ - is the partition function. We can represent this state using the following tensor diagram - Figure 2 (a), where $U = e^{-(\beta/2+it)H}$. For holographic theories, this state is conjectured to be dual to two-sided black hole geometry in anti-de Sitter (AdS) [33] - Figure 3 (a). Hartman and Maldacena [30] studied the entanglement entropy of a subregion $A = A_L \cup A_R$, consisting of two identical subregions: $A_L$ on the left side and $A_R$ on the right. For concreteness, we will consider the case of a finite system with periodic boundary conditions, such that the system is translationally-invariant. Subsystems $A_{L/R}$ are of order half the total system size. For holographic theories, in the leading order in $G_N$, the entanglement entropy $S_{vN}(A_L \cup A_R)$ is equal [31, 32] to the area $\mathscr{A}$ of the extremal surface homologous to $A_L \cup A_R$:

$$S_{vN}(A_L \cup A_R) = \frac{\mathscr{A}}{4G_N} + \mathcal{O}(G_N^0). \tag{4}$$

If there are several competing surfaces then one needs to choose the minimal one. We will define HM surface as:

> **HM surface:** the minimum among the extremal *connected*[b] surfaces homologous to $A_L \cup A_R$.

In this terminology, at early times (times smaller than the system size) the relevant extremal surface is the HM surface $\gamma$ stretching through the black hole interior - Figure 3. Whereas at

---

[b]In $2+1$ bulk dimensions the relevant bulk surface traversing the black hole interior consists of two components. Then we can substitute the word "connected" by "each component connects $A_L$ to $A_R$".

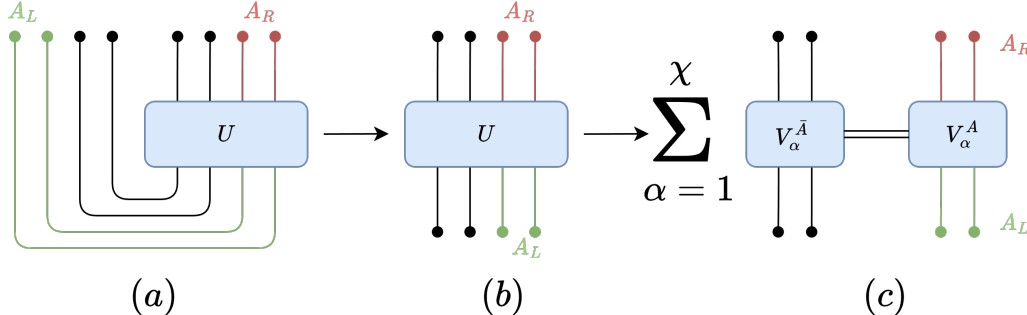

Figure 2: The sequence of transformations from the TFD state to the operator Schmidt decomposition. (a) A conventional TFD state. (b) We bend the legs of the circuit diagram and interpret the left $(L)$ side as the past of the right $(R)$ side. (c) We perform a Schmidt decomposition of the unitary: $V_\alpha^A$ evolves the past subsystem $(A_R)$ into the future subsystem $(A_L)$. Bond index $\alpha$ is responsible for the exchange of information between $A$ and the rest.

late times (times larger than the system size) the minimal extremal surface is *disconnected* $\gamma'$ on Figure 3.

*However, geometrically, HM surface continues to exist and to grow, it is simply not a global minimum anymore. What does its area compute? To answer this question, our main conceptual step is to interpret L and R not as two independent systems, but as one system in two different moments in time.* Algebraically, we "bend" the legs of the tensor diagram on Figure 2 (a) to arrive at Figure 2 (b). Now it is obvious that $A_L$ is in the past $^c$ of $A_R$. In this interpretation, the entanglement entropy of $A$ is the "entanglement in time", as defined in [37], since it is evaluated for a region consisting of a past part $A_L$ and future part $A_R$. *The intuition is that it is sensitive to a connected time-like cut through the evolution operator (HM surface), so it has the information about the circuit depth.* To be more quantitative, we will invoke the notion of operator Schmidt rank.

It is well-known that entanglement entropy is bounded by the logarithm of the density matrix rank. To determine the rank (or approximate rank) we need to cut the operator $U$ *along the time-direction* to see how many non-zero singular values it has - Figure 2 (c).

$$|\text{TFD(t)}\rangle \approx \sum_{\alpha=1}^{\chi} |V_\alpha^{\bar{A}}\rangle |V_\alpha^A\rangle_{A_L \cup A_R}. \tag{5}$$

In other words, we perform a singular value decomposition. Quantity $\chi$ is known as operator Schmidt rank. From this representations it is evident that

$$S_{vN}(A_L \cup A_R) \leq \log(\chi). \tag{6}$$

---

$^c$In the condensed matter literature $S_{vN}(A_L \cup A_R)$ is known as operator entanglement [34–36]. We treat $U$ as an evolution operator - Figure 2 (b), but strictly speaking, $U$ is not unitary. In the Discussion section we discuss how this problem can be easily fixed. Also for high temperatures, it is approximately unitary.

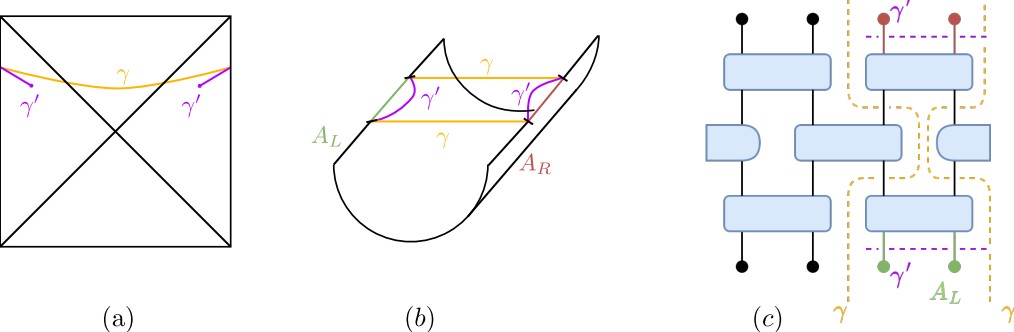

Figure 3: (a) Penrose diagram of AdS black hole. (b) A direct space-time illustration for the cases of $1+2$ dimensional bulk at $t=0$. Semi-circle represents Euclidean state preparation. (c) Quantum brickwork circuit representation with $r=2$. Notice periodic boundary conditions. In all figures $\gamma'$ (purple) is the disconnected saddle. The main conjecture of this paper is that the connected HM surface $\gamma$ (yellow) from (a) and (b) translates to the time-like circuit cut in (c).

This relation is very general: both parts of the inequality do not depend on gate set or whether the system is holographic. It is an important point which we will recall later.

Why is it related to the circuit depth? It is convenient to represent the operator $U$ as a brickwork circuit geometry to emphasize the locality properties - Figure 3 (c) - we can bound the rank by providing *any cut* through the circuit which separates $A_L \cup A_R$ for the rest of the system:

$$\log(\chi) \leq \log D \times (\text{number of cut links}), \tag{7}$$

where $D$ is the local Hilbert space-dimension. One can establish a close parallel between the dominant surfaces in holography and the cuts through the circuit diagram -Figure 3 (c):

- Connected cut $\gamma$ traversing the evolution operator is the direct analogue of the HM surface traversing the black hole interior. It leads to the following bound:

$$\log(\chi) \leq C \times (\text{circuit depth}), \ C = \mathscr{A}_\partial r^2 \log D/4, \tag{8}$$

  where $\mathscr{A}_\partial$ is the area of the boundary of $A_L$ and $r$ reflects the connectivity of the brickwork geometry: how many unitaries in the next layer are connected to a given one at the current layer. For shallow circuits this cut dominates. Hence we have

$$\frac{\mathscr{A}_{HM}}{4G_N} \underbrace{=}_{\text{early time}} S_{vN}(A_L \cup A_R) \leq \log(\chi) \leq C \times (\text{circuit depth}). \tag{9}$$

- A disconnected cut $\gamma'$. In the bulk it corresponds to the union of extremal surfaces for $A_L$ and $A_R$ which stay away from the black hole interior, leading to:

$$\log \chi \leq 2 \log(D)|A_L|. \tag{10}$$

For deep circuits it is this cut that dominates the entanglement entropy:

$$\text{Late times: } \frac{\mathscr{A}_{HM}}{4G_N} \neq S_{vN}(A_L \cup A_R) \leq \log(\chi) \leq 2\log D|A_L| \leq C \times (\text{circuit depth}). \quad (11)$$

However, the HM surface/circuit cut $\gamma$ continue to exist and to grow. The area of the HM surface grows linearly with time and this growth is not expected to change prior to exponentially long (in the system size) times. In principle, there could be an extra island saddle [38–41] appearing but again it is an extra saddle, the HM surface continues to exist even then. Similarly, the circuit depth is not expected to exhibit any qualitative changes prior to exponentially long times. This expectation is based on the fact that for translationally invariant systems the circuit depth is proportional to the number of gates (that is, complexity). Based on that, we conjecture that part of the inequality (9) continues to hold, namely:

$$\frac{\mathscr{A}_{HM}}{4G_N} \leq C \times (\text{circuit depth}). \quad (12)$$

# 3  Discussion

This short note was dedicated to the measures of complexity of an evolution operator $U = e^{(-it-\beta/2)H}$. We argued that for holographic systems, the area of HM surface (in units of $4G_N$) bounds from below the number of layers in a quantum circuit - eq. (12). Our arguments are based on rigorously establishing the following bound at early times:

$$\frac{\mathscr{A}_{HM}}{4G_N} \leq \log \chi \leq C \times (\text{circuit depth}), \quad (13)$$

And then extrapolating to later times based on the fact that the leftmost quantity ($\mathscr{A}_{HM}$) and the rightmost (circuit depth) continue to grow at a steady rate, unlike $\log \chi$. The arguments in this paper can be improved: we treated holographic conformal field theories as finite dimensional systems and we obtained only a lower bound on complexity, rather than showing that it is *equal* to some bulk quantity. Also we tacitly assumed translation-invariance, otherwise it is not clear how to define the circuit depth and $\mathscr{A}_{HM}$ depends on the choice of a subsystem: currently we do have $\mathscr{A}_\partial$ in our formulas, but it will conveniently cancel out with the same factor in $\mathscr{A}_{HM}$ because the HM surface preserves the shape of the boundary region through its bulk journey [30]. *However, our arguments have provided a microscopic evidence that a black hole interior size captures quantum circuit complexity.* Let us offer some closing remarks about the potential lessons from this relation.

**From circuit depth to complexity.** The relation to the area of an extremal surface in the interior seem to favor "complexity=volume", rather than "complexity=action". For translationally invariant systems, quantum gate complexity is equal to the depth times the system size. Similarly, if we sum (integrate) over all codimension-two surfaces we will get a measure of volume (however it is not obvious it would coincide with the conventional volume), hence we very roughly have:

$$(\text{volume}) \lesssim C \times (\text{gate complexity}). \quad (14)$$

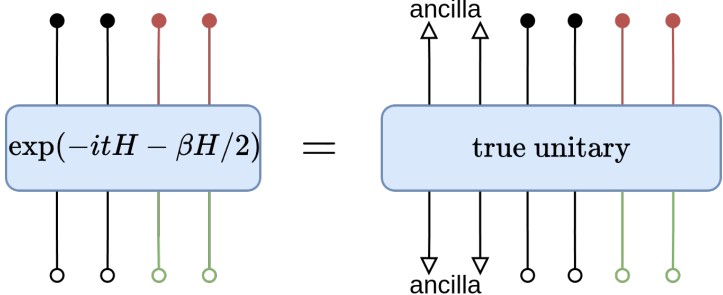

Figure 4: Illustration of the Sz.-Nagy dilation.

**Gate set and minimality.** It is important to emphasize that $\log \chi$ is an intrinsic property of $U$, it does not depend on gate decomposition. Whereas the upper bound on $\log \chi$ can be obtained from *any* circuit. Hence we are bound to conclude that $\mathscr{A}_{HM}/4G_N$, bounds from below $C \times (\text{circuit depth})$ for any circuit - we are free to choose any gate set, any connectivity and search for the minimal circuit representation globally.

**Properties of the operator Schmidt rank.** In general, $\log \chi$ is greater than the operator entanglement entropy $S_{vN}(A_L \cup A_R)$. However, for Clifford circuits the entanglement spectrum is actually flat, so they are equal[d].

Measures of circuit complexity are supposed to satisfy two simple properties: subadditivity and switchback [10]. In Appendix A we argue that the log of operator rank satisfies them too:

- Subadditivity:
$$\log \chi(U_1 U_2) \leq \log \chi(U_1) + \log \chi(U_2). \tag{15}$$

- Switchback: for a local unitary operator $W$ and a scrambling unitary $U$ the following holds for large enough times:
$$\log \chi(U W U^{\dagger}) \leq 2 \log \chi(U) - 2\delta, \ \delta < t_* \tag{16}$$

The actual switchback effect predicts that the above relation is actually equality, but we will be able only to demonstrate inequality.

**A comment on non-unitarity and UV-cutoff.** So far we only used the word "operator" for $U = e^{(-it-\beta/2)H}$ because it is not a unitary. Hence, strictly speaking, we cannot use the word circuit depth for $U$. Fortunately, we can represent $U$ as a truly unitary operator at the expense of introducing ancilla system and projecting it on a given state - Figure 4. This statement reflects the mathematical fact that a non-unitary matrix can be embedded as a block into a

---

[d]We are grateful to Zixia Wei for pointing this out.

unitary matrix - sometimes it is referred to as Sz.-Nagy dilation theorem. For simple cases such embedding can be found explicitly [42].

The purpose of this projection is to remove UV degrees of freedom. One can imagine separating in the circuit the non-unitary part $e^{-\beta H/2}$ from the unitary $e^{-iHt}$. The former gives a constant overhead to the circuit depth, whereas the latter grows linearly in time $\sim t/\varepsilon$, where $\varepsilon$ is the UV cutoff. Naively, the bound on circuit depth in terms of $\mathscr{A}_{HM}/4G_N$ then becomes very loose, because in the area, $\varepsilon$ enters only as a constant shift. However, the idea is that $e^{-\beta H/2}$ removes the UV-degrees of freedom from the system, hence $e^{-iHt}$ acts only within the low-energy subspace, effectively reducing the cutoff to $\beta$. Then the area and the depth become comparable, both growing as $t/\beta$.

# Acknowledgment

We would like to thank Thomas Hartman, Juan Maldacena, John Preskill, Zixia Wei, Ying Zhao for comments.

AM acknowledges funding provided by the Simons Foundation, the DOE QuantISED program (DE-SC0018407), and the Air Force Office of Scientific Research (FA9550-19-1-0360). The Institute for Quantum Information and Matter is an NSF Physics Frontiers Center. AM was also supported by the Simons Foundation under grant 376205.

# A   Properties of the operator-rank

**Subadditivity:**   This reflects a simple fact that unitaries $U_1, U_2$ might partially cancel each other. Assuming that both $U_1$ and $U_2$ have the exact operator ranks $\chi_{1,2}$ (with respect to the decomposition into $A$ and $\overline{A}$), the operator rank of $U_1 U_2$ is less than the product of the ranks.

In the approximate setup this statement is less obvious and it would be interesting to analyze it. We can define $\chi_\varepsilon$ as the operator rank of the matrix $U_\varepsilon$ which approximates $U$ with accuracy $\varepsilon$: $||U - U_\varepsilon|| \le \varepsilon$. Then the matrix $U_{1,\varepsilon} U_{2,\varepsilon}$ has the rank less than $\chi_{1,\varepsilon} \chi_{2,\varepsilon}$, but unfortunately its distance to $U_1 U_2$ is only bounded by $2\varepsilon$:

$$||U_{1,\varepsilon} U_{2,\varepsilon} - U_1 U_2|| = ||U_{1,\varepsilon} U_{2,\varepsilon} - U_{1,\varepsilon} U_2 + U_{1,\varepsilon} U_2 - U_1 U_2|| \le ||U_{1,\varepsilon} U_{2,\varepsilon} - U_{1,\varepsilon} U_2|| +$$
$$+ ||U_{1,\varepsilon} U_2 - U_1 U_2|| \le ||U_{1,\varepsilon}|| ||U_{2,\varepsilon} - U_2|| + ||U_{1,\varepsilon} - U_1|| ||U_2|| \le 2\varepsilon. \tag{17}$$

Here $||\cdot||$ is any operator norm that is unitary invariant:

$$||UMV|| = ||M||, \text{ for unitary } U, V, \tag{18}$$

and subadditive. For example, any Schatten $p$-norm. Unfortunately, subadditivity (as a function of unitary with a given bi-partition) does not hold for operator entanglement, one can easily construct numerical counter-examples to $S_{vN}(A|U_1 U_2) \le S_{vN}(A|U_1) + S_{vN}(A|U_2)$ and also to $|S_{vN}(A|U_1) - S_{vN}(A|U_2)| \le S_{vN}(A|U_1 U_2)$.

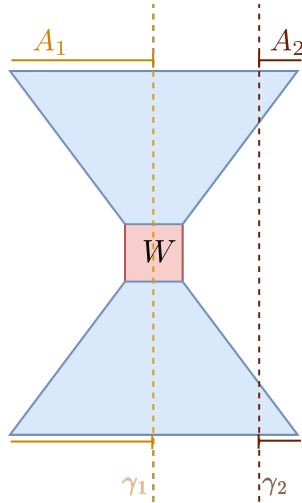

Figure 5: Illustration of the switchback effect. Blue denotes the collection of unitaries forming $U$ and $U^\dagger$. The hour-glass shape illustrates the cancellations around $W$.

**Switchback:** This effect reflects the fact that $U$ and $U^\dagger$ partially cancel each other only around $W$. This is illustrated by Figure 5. So far we could ignore the fact that $\chi$ depends on the choice of subsystem $A$ because we could concentrate on the translationally-invariant case. The unitary corresponding to the switchback effect manifestly breaks this invariance. For example, for the subsystem $A_1$ we expect that $\log \chi(UWU^\dagger) \approx 2\log \chi(U)$ (no switchback), whereas for $A_2$ we can use the cut $\gamma_2$ to definitely say that $\log \chi(UWU^\dagger) \leq 2\log \chi(U) - 2\delta$, where $\delta$ depends on the precise location of $A_2$. However, $\delta$ cannot be bigger than $t_*$, where time $t_*$ is require to scramble the operator $W$ - because past the time $t_*$ the circuit does not have cancellations anymore.

For unitaries of the form $UWU^\dagger$, the operator entanglement is sometimes referred to as a local operator entanglement [43, 44].

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
