# Peer review of "From black hole interior to quantum complexity through operator rank"

_SciPost Physics_

## Round 2 · Referee Report · Anonymous (Referee 1) · 2025-6-2

Strengths

I think this paper’s strengths include its novelty and brevity. I’ve worked in this field for quite a while and found the ideas put forward here fresh and stimulating. The discussion prompted many interesting questions for me while reading. I also appreciated the to-the-point presentation.

Weaknesses

I think this paper’s weaknesses include some underdeveloped notions. For example, what does the approximation sign in Eq 5 mean? Are there issues in defining chi in a system with continuous time evolution? How does the degree of approximation, for example as discussed in the appendix, enter into chi?

Report

I think this paper provides fresh insight into a fairly mature subject. I largely agree with the early time analysis, modulo a point I will mention momentarily. My immediate intuition for the longer time conjecture (after the HM surface is no longer dominant) is that it is reasonable in that it draws on the same kind of circuit and tensor network analogies underlying other complexity/geometry conjectures. Moreover, the switchback effect is an important test of such conjectures. I think this is an interesting and non-trivial step in the literature and it could reasonably be published in SciPost.

Requested changes

However, I recommend the author address the following points before a final decision:

I think the paper needs to be more explicit about various definitions. For example, what does the approximation sign in Eq 5 mean? How precisely is r defined? One concern is that, without a proper regularization, the operator rank might be immediately maximal. This would not invalidate the early time bounds, but the upper bound on the entanglement would be quite loose and the lower bound on complexity would be artificially high since we are requiring too exact a version of the unitary.

I think the paper should delineate more clearly which situations this conjecture is expected to apply to. The main analysis considers time-evolved thermofield double states. Moreover, since the switchback effect is discussed, it must be meant to apply to evolutions involving shocks as well. However, what about Euclidean insertions, for which the geometry may have a Python’s lunch (this is incidentally related to the proposal for dealing with non-unitarity in Fig 4)? Here there is a potential breakdown between depth and complexity since these situations are thought to involve an expensive post-selection process (or amplitude amplification process).

The points mentioned above will help clarify the precise definition and scope of the conjecture and so deserve to be addressed. If these are addressed, I think the work is sufficiently novel and stimulating to meet SciPost’s requirements.

Recommendation

Ask for minor revision

---

## Round 2 · Referee Report · Anonymous (Referee 2) · 2025-6-13

Strengths

Original ideas, and tries to make complexity=geometry sharp.

Weaknesses

Several points are sloppy.

Report

This is an interesting short paper that aims at making the connection between complexity of the boundary state and some geometric features of the bulk sharp. This is nice since most of the proposals for complexity are just conjectures are have not been proven. Here, the author introduces a bound between the area of a surface in the bulk, first discussed by Hartman and Maldacena, and a property of the CFT preparation of the state. The authors says that the bound is rigorous at early times, but could potentially be extrapolated at late times.

While the ideas of the paper are interesting, and I do appreciate that the paper is short, so should be seen more as a "proof of concept" type of paper, there are still too many details that are sloppy for me to recommend the paper be published in its current form. I list the changes requested below.

Requested changes

  1. The operator $U=e^{-\beta H/2 + i t H}$ is NOT a unitary. The authors does end up saying this at the very end of the paper, but this is needed much earlier. At the very least, there should be a comment there and more detailed explanations can be given later.

  2. What is the meaning of the approximate sign in equation 5? If this is a singular value decomposition, then there is an equal sign. My guess is that perhaps it is precisely because the matrix is not a unitary? In any case, this should be specified.

  3. This point is important: the author claims to DERIVE a bound at early times, but for this treats the holographic CFT as a discrete gate system. This cannot be done so simply. One of the major challenges of complexity=volume is that the CFT is a field theory, and things are plagued by UV divergences. So if the author wants to claim any result be derived on area terms, etc. First, a detailed explanation of the regularization scheme should be given, and then, one can see if the proof follows. Alternatively, one can apply these ideas to systems like perfect or random tensors, but then we are dealing with toy models of holography and NOT with true holographic systems. It is crucial the author address this point.

Recommendation

Ask for major revision

---

## Editorial Decision

awaiting_resubmission